**Data Availability Statement:** All relevant data are within the paper and its Supporting information files.

**Funding:** We confirm that the authors received no specific funding for this work.

# Nutritional assessment and associated factors in children with congenital heart disease—Ethiopia

**Temesgen Tsega**[1]*, **Tigist Tesfaye**[2], **Azene Dessie**[3], **Tesfalem Teshome**[4]

**1** Department of pediatrics and child health, Saint Paul hospital millennium Medical college, Addis Ababa, Ethiopia, **2** Saint Peter medical college, Addis Ababa, Ethiopia, **3** Children Cardiac Center, Addis Ababa, Ethiopia, **4** Department of public health, Saint Paul hospital millennium Medical college, Addis Ababa, Ethiopia

* yafettemu@gmail.com

## Abstract

### Introduction

Worldwide, congenital heart disease is the principal heart disease in children and constitutes one of the major causes of infant mortality, particularly in developing countries. Infants and children with congenital heart disease exhibit a range of delays in weight gain and growth. In some instances, the delay can be relatively mild, whereas in other cases, cause the failure to thrive.

### Objectives

To determine the nutritional status and associated factors of pediatric patients with congenital heart disease.

### Material and method

A cross sectional analytical study conducted over a period of 6 months (Feb to Jul 2020). A total of 228 subjects with congenital heart disease who visited the cardiac center during the study period where included until the calculated sample size attained. Data is collected from patient's card and their care giver. Data was then analyzed using Statistical Package for Social Sciences (SPSS) for windows version 25.0. Odds Ratio with 95% Confidence Interval (CI) was used to determine the effect of the independent variables on the outcome variable and P-value less than 0.05 was considered statistically significant.

### Results

A total of 228 children ranging from 3month to 17yrs of age with mean age of 4.7 years (SD = 3.8 years) were included in the study. Most of the subjects had acyanotic heart disease accounting for 87.7%. The overall prevalence of wasting, underweight and stunting were 41.3%, 49.1% and 43% respectively. Children with congenital heart disease and having pulmonary hypertension, were found more likely to develop wasting compared to those without pulmonary hypertension with an odds of 1.9 (95% CI: 1.0–3.4) and also have greater

**Competing interests:** The authors have declared that no competing interests exist.

**Abbreviations:** AS, Aortic Stenosis; ASD, Atrial Septal Defect; AVSD, AtrioVentricular Septal Defect; CCC, Children's Cardiac Center; CHD, Congenital Heart Disease; CHF, Congestive Heart Disease; CoA, Coarctation of Aorta; EMDHS, Ethiopia Mini Demographic and Health Survey; OPD, Out Patient Department; PA, Pulmonary Atresia; PAH, Pulmonary Arterial Hypertension; PDA, Patent Ductus Arteriosus; PS, Pulmonary Stenosis; SPMMC, St. Paul Hospital Millennium Medical College; SPSS, Statistical Package for Social Science; TA, Truncus Arteriosus; TAPVR, Total Anomalous Pulmonary Venous Return; TGA, Trans-position of Great Arteries; TOF, Tetralogy Of Fallot; VSD, Ventricular Septal Defect; WHO, World Health Organization.

chance of stunting with an odds of 1.9 (95% CI: 1.0–3.4). Children 5 to 10 years of age were 2.3 times more likely to be underweight.

## Conclusion

Malnutrition is a major problem in pediatric patients with congenital heart disease. Pulmonary hypertension and older age are associated with increased risk of undernutrition.

# 1. Introduction

## 1.1. Background

Congenital heart disease (CHD) is a defect in the heart or major blood vessels that are present in children at birth & occurs in approximately 1% of live births. It is usually defined as clinically significant structural heart disease present at birth [1,2].

Worldwide, CHD is the principal heart diseases in children and constitutes one of the major causes of infant mortality, particularly in developing countries [3]. Infants and children with CHD exhibit a range of delays in weight gain and growth. In some instances, the delay can be relatively mild, whereas in other cases, it could be severe causing failure to thrive [4].

Good nutrition is essential for survival, physical growth, mental development, performance, productivity, health and well-being across the entire life span from the earliest stages of fetal development, at birth, and through infancy, childhood, adolescence and on into adulthood [4]. Infants and children are more likely to suffer from poor nutrition than compared to adults. The first reason is newborn infants have low stores of fat and protein. The smaller the child, the fewer reserves of energy they have. This means that they can only cope with starvation for shortened periods. The second reason is high nutritional demands for growth: The amount of nutrition children require is greatest during infancy, because of their rapid growth during this period. The third reason is rapid development in the nervous system: the child's brain grows rapidly during the last four months of pregnancy and also during the first two years of the life. The connections between the nerve cells in the brain are being formed during this time therefore good nutrition is important to ensure that this occurs properly. The fourth reason is Illness: the child's nutrition may be compromised following an episode of illness or surgery. The body's energy requirements are increased; thus intake of food and nutrients should be increased [4–6].

Prevalence of growth failure is estimated to be 64% in children with CHD living in developed countries while this number likely to be higher in those living in developing countries for the reason than malnutrition perse is much more common because of other additional factors.

Multiple factors are mentioned in literatures to answer why would patients with CHD develop malnutrition. These are chromosomal abnormalities, feeding problems causing inadequate nutrition and malabsorption that occurs because of edematous gastrointestinal system in those with chronic heart failure. Chronic cyanosis and heart failure dysfunction body metabolism and the susceptibility to infection again will result higher body metabolism. Those children with CHD and undernutrition have worse prognosis implicated by poor somatic growth, repetitive admissions, unfavorable outcome after intervention and finally they will succumb to death [6].

Anthropometry means the study of human body measurements in comparative bases. It utilizes primarily indices of growth which includes weight, stature (length/height) and head

circumference especially for younger children. Triceps skinfold thickness, subscapular skinfold thickness, and mid-upper arm circumference are secondary measures to estimate the body composition [4]. Percentile curves are used to compare the measured values to normal ranges of population data.

## 1.2. Statement of the problem

Significant percentage of undernutrition and short stature seen in children with congenital heart disease. The presence of malnutrition will prone them for infection and the prognosis is grave even if correction is done later in life. The chance of developing malnutrition increases in those having CHD with cyanosis, multiple heart defects, heart failure, delayed intervention, anemia and pulmonary hypertension. Because of multiple reasons children with CHD in developing countries are not getting the opportunity for corrective intervention which indirectly makes them susceptible to develop malnutrition [7–12].

For the above-mentioned reasons, risk factors and associated comorbidities, malnutrition is a common finding we see in our patients with CHD, complicating the course and outcome of the disease. Despite the significant morbidity and mortality associated with undernutrition in CHD, very little emphasis has been given in the management and prevention of this complication in our clinical practice.

## 1.3. Significance of the study

The conclusion obtained from such study will help health professionals to emphasis on nutritional assessment of patients with CHD and respective measures either to prevent it or early intervention to avoid additional complications. The results of this study can also be an input to develop nutritional guidelines for infants & children with congenital heart disease to provide adequate calories and protein, considering potentially increased needs, and promote optimal weight gain and growth velocity. Such studies in general are useful for policy makers to have an insight about patients with comorbidities like congenital heart disease are at increased risk of such complication in addition to the primary disease itself.

## 2. Methodology

### 2.1. Study area

This study was conducted at pediatrics cardiac clinic of cardiac center—Ethiopia.

### 2.2. Study design and period

A cross sectional, prospective, analytic study was conducted over a period of 6months (February 2020 to July 2020).

### 2.3. Source and Study population

**2.3.1. Source population.** All pediatric patients with congenital heart disease age 0–18yrs seen at outpatient clinic of children's cardiac center.

**2.3.2. Study population.** All pediatric patients with congenital heart disease age 0–18yrs seen at outpatient clinic of children's cardiac center during the study period.

### 2.4. Inclusion criteria

All pediatric patients age 0–18yrs with congenital heart disease seen at outpatient clinic of children's cardiac clinic who didn't undergone intervention.

## 2.5. Exclusion criteria

All patients with risk factors other than CHD that contribute to malnutrition like genetic disorders, chronic illness, prematurity and low birth weight were excluded.

## 2.6. Sample size determination and sampling technique

The sample size was determined using the single population proportion formula, taking prevalence (P) of 84% from a research done in Egypt (13), Z = 1.96, and assuming a 10% non-response rate, giving the total sample size to be **228**.

Study subjects fulfilling the inclusion criteria were included consecutively until sample size is reached.

## 2.7. Data collection

Data collection was conducted by a trained general practitioner using a structured data inquiry sheet developed by the primary investigator. The data inquiry sheet included demographic characteristics including age, gender, age of diagnosis, social status of the parents and Other data including cardiac diagnosis at echocardiography, presence of pulmonary hypertension, weight, height, length, and body mass index. Relevant laboratory result like hemoglobin was included.

## 2.8. Method of data collection, tool and personnel

A convenient sampling technique was used to include subjects that fulfilled the inclusion criteria that visited the cardiac clinic at the time of data collection until the sample size was achieved.

Weight was measured by a single person with the same weight scale.

Height was measured using a tape meter when the patient is lying in supine position flat on a rigid surface for those below two years and for those above two years who can't stand.

WHO Z- score classification for malnutrition is used to assess and categorize the nutritional status of these children included in our study. Acute malnutrition assessed by weight/length score, chronic malnutrition assessed by length/age score, while poor nutrition assessed by weight/age score.

## 2.9. Data processing and analysis

Interpretation of anthropometric values was based on the WHO Z-scores, interpreted as moderate and severe wasting if weight for height is $< -2SD$ to $> -3SD$ and $< -3SD$ respectively.

Moderately and severely underweight if weight for age is $< -2SD$ to $> -3SD$ and $< -3SD$ respectively.

Moderately and severely stunted if length/height for age was $< -2SD$ to $> -3SD$ and $< -3SD$ respectively.

Data analysis was done using SPSS version 25 statistical package. Individual questionnaire was checked before data entry into the software. Further data cleaning was performed to check for outliers, missed values and any inconsistencies before the data were analyzed using the software.

Pearson's chi square and fisher-exact test was used to find the association between categorical variables and For variables with p-value less than 0.20 in univariable logistic regression, multivariable logistic regression analyses were conducted to control the cofounders and to assess the association between independent variables and nutritional status. A p-value of $\leq 0.05$ was considered significant.

## 2.10. Ethical consideration

The study was approved by Institutional research board (IRB) of SPHMMC according to Ethiopian national research guideline. The privacy and confidentiality of all participants was secured, and informed written consent was taken from all study subjects and/or their care givers.

# 3. Results

A total of 228 children with congenital heart disease were included in the research. The mean age was 4.7years (SD 3.8yrs) and range of 3months to 17yrs (see Table 1).

Majority of the study subjects have acyanotic congenital heart disease (200/228) accounting for 87.7%. The age at diagnosis for most of the patients, 93.4% (213), is after 12month. (See Table 2).

Overall prevalence of underweight, wasting and stunting among CHD patients was 49.1% (112), 41.3% (94) and 43% (98), respectively.

The commonest congenital heart disease was VSD, occurring alone in 25% and co-exising with other congenital heart defects in 40.8%, followed by PDA (15.4%). The commonest cyanotic CHD is TOF occurring in 9.2% of all the CHDs. See Table 3.

From the 200 study subjects with acyanotic CHD, burden of underweight was 48%, wasting was 41.5% and stunting was 42.5%.

In the cyanotic CHD group, 53% of them were underweight, 35.7% were wasted and 42.8% of them were stunted. (see Table 4).

Chi-square test was done to check if there is significant difference between CHD patients with PAH and without PAH based on their nutritional status using $P<0.05$ as significant.

Among Acyanotic patients, PAH looks to have some degree of association with all underweight, wasting and stunting but no significant difference was observed between the two groups (acyanotic vs cyanotic).

Among cyanotic patients, PAH is found to have association with stunting but no significant difference was observed between patients with and without PAH based on their nutritional status.

Bivariate analysis was done to check if; Patient's sex, age, residence, age at diagnosis, acyanotic CHD, VSD, ASD, PDA, cyanotic CHD and TOF, have significant association with underweight at $P<0.2$. Multivariable analysis was done to see if there is significant association of factors identified on bivariate analysis with outcome variable at $P<0.05$.

Children between 5 and 10 years of age were 2.3 times more likely to be underweight than those between the age of 1 and 3 years.

Children with ASD were 70% less likely to be underweight compared to others with CHD but without ASD 70% (95% CI:0.1–0.9) (see Table 5a).

Bivariate analysis was done to check if; Patient sex, age, residence, age at diagnosis, Acyanotic CHD, VSD, ASD, PDA, PAH, cyanotic CHD and TOF, have significant association with wasting at $P<0.2$. Multivariable analysis was done to see if there is significant association of factors identified on bivariate analysis with outcome variable at $P<0.05$. We are mentioning here only those variables which showed associations.

As described in Table 5b, children with PAH are more likely to have wasting compared to those without PAH with an odds of 1.9 (95% CI: 1.0–3.4) among children with CHD.

Bivariate analysis was done to check if; Patient sex, age, residence, age at diagnosis, Acyanotic CHD, VSD, ASD, PDA, cyanotic CHD and TOF, have significant association with stunting at $P<0.2$. Multivariable analysis was done to see if there is significant association of factors identified on bivariate analysis with outcome variable at $P<0.05$.

**Table 1. Socio-demographic characteristics of the study participants at the cardiac center Ethiopia.**

| Variables | Frequency | Percent |
|---|---|---|
| **Age group** | | |
| <1 years | 35 | 15.3 |
| 1–3 years | 73 | 31.9 |
| 3–5 years | 44 | 19.2 |
| 5–10 years | 74 | 32.4 |
| >10 years | 2 | 0.9 |
| **Gender** | | |
| Male | 102 | 44.7 |
| Female | 126 | 55.3 |
| **Residence** | | |
| Urban | 105 | 46.1 |
| Rural | 123 | 53.9 |
| **Religion** | | |
| Orthodox | 130 | 57.0 |
| Muslim | 74 | 32.5 |
| Protestant | 23 | 10.1 |
| Catholic | 1 | 0.4 |
| **School level** | | |
| PKG | 174 | 76.3 |
| Grade 1–4 | 45 | 19.7 |
| Grade 5–8 | 7 | 3.1 |
| Grade 9–12 | 2 | 0.9 |
| **Parental Marital status** | | |
| Single | 1 | 0.4 |
| Married | 225 | 98.7 |
| Widowed | 2 | 0.9 |
| **Parental Education level** | | |
| Can't Read or Write | 16 | 7.0 |
| Read and Write | 18 | 7.9 |
| Below high school | 84 | 36.8 |
| Complete high school | 48 | 21.1 |
| College graduate | 62 | 27.2 |
| **Occupation** | | |
| Farmer | 41 | 18.0 |
| Government employee | 54 | 23.7 |
| Daily laborer | 31 | 13.6 |
| Merchant | 19 | 8.3 |
| Private employee | 83 | 36.4 |
| **Monthly income** | | |
| <1000 birr | 38 | 16.7 |
| 1001–3000 birr | 109 | 47.8 |
| 3001–5000 birr | 42 | 18.4 |
| >5000 birr | 39 | 17.1 |

Children with ASD have decreased chance of being stunted by 70% (95% CI: 0.1–0.9) compared to those with CHD but without ASD.

Children with PAH are more likely to be stunted compared to those without PAH with an odds of 1.9 (95% CI: 1.0–3.4) among children with CHD (see Table 5c).

**Table 2. Clinical characteristics of CHD patients.**

| Variables | Frequency | Percent |
|---|---|---|
| **CHD type** | | |
| Acyanotic | 200 | 87.7 |
| Cyanotic | 28 | 12.3 |
| **Age at diagnosis** | | |
| <01 month | 15 | 6.6 |
| 01–12 months | 58 | 25.4 |
| >12 months | 155 | 68.0 |
| **Pulmonary hypertension** | | |
| Yes | 68 | 29.8 |
| No | 159 | 69.7 |
| **Anemia** | | |
| Yes | 34 | 14.9 |
| No | 194 | 85.1 |
| **Underweight** | | |
| No underweight | 116 | 50.8 |
| Moderate | 49 | 21.5 |
| Severe | 63 | 27.6 |
| **Wasting** | | |
| No wasting | 134 | 58.8 |
| Moderate | 38 | 16.7 |
| Severe | 56 | 24.6 |
| **Stunting** | | |
| No stunting | 130 | 57 |
| Moderate | 49 | 21.5 |
| Severe | 49 | 21.5 |

## 4. Discussion

Children with CHD usually do not exhibit failure to thrive at the time of birth and during neonatal period unless the CHD is hemodynamically significant during that period. Otherwise, it usually become apparent after few weeks of like when the pulmonary pressure declined to its nadir. The severity and degree of malnutrition rely on mainly additional factors which characterizes the CHD. These are presence of cyanosis, development of heart failure and pulmonary hypertension [13–15].

Indian study of anthropometric data in children with CHD revealed dietary intake, educational level, occupational and socioeconomic status did not affect the chance of being malnourished. In low- and middle-income countries, the prevalence of abnormal preoperative anthropometry is high attributed to late presentation, delays in corrective intervention, and frequent hospitalizations related to respiratory infections [6].

In this study, out of 228 participants the overall burden of underweight was 49.1% (112). Among the acyanotic group 48% (96) were underweight which is lower than the cyanotic group where prevalence reaches 53% [16]. In a similar study done in Uganda, involving 194 children with CHD, underweight assessed for those 0-10years, accounted for 42.5%. [7]. Oday Faris Washeel and his colleagues found out underweight accounted for 32% of the 65 studied subjects, 0-5years with CHD who visited the heart center in Nursing College, Al-Muthanna University, Samawah, Iraq. [5]. On the other hand, an Indian study to identify determinants of

**Table 3. Type and prevalence of cardiac defects among CHD patients.**

| Types of CHD | Frequency | Percent |
|---|---|---|
| **Acyanotic** | | |
| VSD | 57 | 25.0 |
| PDA | 35 | 15.4 |
| ASD | 22 | 9.6 |
| AVSD | 15 | 6.6 |
| CoA | 1 | 0.4 |
| PS | 12 | 5.3 |
| AS | 10 | 4.4 |
| VSD+ASD | 7 | 3.1 |
| VSD+PDA | 15 | 6.6 |
| VSD+PDA+CoA | 6 | 2.6 |
| AVSD+PDA | 4 | 1.8 |
| PS+VSD | 5 | 2.2 |
| AS+PDA | 9 | 3.9 |
| **Cyanotic** | | |
| TOF | 16 | 7.0 |
| TOF+PA+PDA | 6 | 2.2 |
| PA+PDA | 2 | 0.9 |
| TGA+VSD | 3 | 1.3 |
| TAPVR | 1 | 0.4 |

malnutrition in children with congenital heart disease (CHD) which involved 476 pre-operative patients, found a higher underweight percentage, 59%. [6].

The overall prevalence of wasting in this study was 41.3% (94). Children with acyanotic CHD were observed to have a higher percentage of wasting as compared to those with cyanotic CHD, accounting 41% and 35%, respectively. This percentage is higher in the Indian and Ugandan study where wasting was detected in 56% and 58% of the children with CHD respectively [6,7]. An even higher percentage was reported from the study in Al-Muthanna University, Samawah, Iraq, 64%, which was attributed to premature delivery and low birth weight [5]. Infants delivered premature and with low birth weight were excluded from this study.

**Table 4. Comparison of CHD patients with and without pulmonary hypertension based on their nutritional status.**

| Variables | APAH | AWTPAH | $X^2$ (P value) | CPAH | CWTPAH | $X^2$ (P value) |
|---|---|---|---|---|---|---|
| **Underweight** | | | | | | |
| No | 27 | 76 | 5.7 (0.057) | 0 | 13 | 1.9 (0.17) |
| Yes | 39 | 57 | | 2 | 14 | |
| **Wasting** | | | | | | |
| No | 33 | 83 | 4.2 (0.12) | 1 | 17 | 0.2 (0.66) |
| Yes | 33 | 50 | | 2 | 9 | |
| **Stunting** | | | | | | |
| No | 31 | 83 | 5.6 (0.06) | 0 | 17 | 2.9 (0.09) |
| Yes | 35 | 50 | | 2 | 10 | |

APAH = Acyanotic with pulmonary hypertension, AWTPAH = Acyanotic without pulmonary hypertension, CPAH = Cyanotic with pulmonary hypertension, CWTPAH = Cyanotic without pulmonary hypertension, $X^2$ = Chi square.

**Table 5.**   a. Bivariate and multivariate analysis for Underweight. b. Bivariate and multivariate analysis for wasting. c. Bivariate and multivariate analysis for stunting.

| Variables | Underweight (Yes) | Normal weight | COR (95% CI) | P Value | AOR (95% CI) | P Value |
|---|---|---|---|---|---|---|
| Residence | | | | | | |
| Rural | 59 | 46 | 1.7 (1.0–2.9) | 0.05* | 1.6 (0.9–2.8) | 0.11 |
| Urban | 53 | 70 | 1.0 | | 1.0 | |
| Age | | | | | | |
| <1 yr | 27 | 46 | 2.0 (0.9–4.6) | 0.09* | 2.3 (0.9–5.3) | 0.053 |
| 1–3 yr | 19 | 16 | 1.0 | | 1.0 | |
| 3–5 yr | 21 | 23 | 1.6 (0.7–3.3) | 0.3 | 1.6 (0.7–3.5) | 0.2 |
| 5–10 yr | 44 | 30 | 2.5 (1.3–4.8) | 0.01* | 2.3 (1.1–4.5) | 0.02* |
| >10 yr | 1 | 1 | 1.0 | | 1.0 | |
| ASD | | | | | | |
| Yes | 6 | 16 | 0.4 (0.1–0.9) | 0.04* | 0.3 (0.1–0.9) | 0.03* |
| No | 106 | 100 | 1.0 | | 1.0 | |
| Variables | Wasting (Yes) | Wasting (No) | COR (95% CI) | P Value | AOR (95% CI) | P Value |
| VSD | | | | | | |
| Yes | 19 | 38 | 0.6 (0.3–1.0) | 0.16* | 0.6 (0.3–1.2) | 0.14 |
| No | 75 | 96 | 1.0 | | 1.0 | |
| PDA | | | | | | |
| Yes | 19 | 16 | 1.9 (0.9–3.9) | 0.09* | 1.7 (0.8–3.5) | 0.19 |
| No | 75 | 118 | 1.0 | | 1.0 | |
| PAH | | | | | | |
| Yes | 34 | 34 | 1.7 (0.9–2.9) | 0.08* | 1.9 (1.0–3.4) | 0.04* |
| No | 60 | 100 | 1.0 | | 1.0 | |
| Variables | Stunting (Yes) | Stunting (No) | COR (95% CI) | P Value | AOR (95% CI) | P Value |
| Residence | | | | | | |
| Rural | 53 | 52 | 1.8 (1.0–3.0) | 0.04* | 1.7 (0.9–2.9) | 0.06 |
| Urban | 45 | 78 | 1.0 | | 1.0 | |
| ASD | | | | | | |
| Yes | 5 | 17 | 0.4 (0.1–1.0) | 0.05* | 0.3 (0.1–0.9) | 0.04* |
| No | 93 | 113 | 1.0 | | 1.0 | |
| PAH | | | | | | |
| Yes | 37 | 31 | 1.9 (1.1–3.4) | 0.02* | 1.9 (1.0–3.4) | 0.03* |
| No | 61 | 99 | | | | |

Ninety seven out of the 228 study subjects were found to be stunted which equals 43% (98) and no significant difference was seen between the cyanotic and acyanotic CHD. Comparable result was seen in the Ugandan study with stunting of 45%. [7].

In this study pulmonary hypertension is associated with increased risk of being wasted (OR 1.9 (95% CI: 1.0–3.4)) and stunted (OR 1.9 (95% CI: (1.0–3.4)). Acyanotic patients with PAH looks to have some degree of association with all underweight, wasting and stunting but no significant difference on the risk of malnutrition between acyanotic and cyanotic CHD. But in general children with PAH are more likely to have wasting and stunting compared to those without PAH.

In contrast to ours, findings from the study done by Ijeoma Arodiwe and his colleagues stated that severe malnutrition and stunting were seen more in those with cyanotic CHD and developed pulmonary hypertension in comparison to acyanotic heart disease who were only wasted [4]. In another study by Varan B, Tokel K and Yilmaz G. to compare the development

of malnutrition in cyanotic and acyanotic ones in the presence and absence of pulmonary hypertension, they found out those with cyanotic CHD are likely to develop stunting than wasting, and out of which who developed pulmonary hypertension are severely affected [16].

Of the total patients 34 (14.9%) of them had anemia with variable degree of severity from mild to severe and it was not found to have association with increased risk of malnutrition in this study; but in Ugandan study they found out association and those children are with moderate or severe anemia [7,17].

The other finding in this study is the decreased risk of being underweight and stunted (70% (95% CI: 0.1–0.9)) in children with ASD than those CHD without ASD.

In the Indian study, they concluded, among many other risk factors identified, older age at surgical correction of the CHD was associated with increased risk of malnutrition and poor recovery after surgery [6]. Similar conclusion was made in the Nigerian study where they found Children in the older age group and those presenting late are more prone to malnutrition and poor growth [4]. The finding in this study was also similar to the above findings. It was found that children above 5 years (especially 5 to 10 years of age) are 2.3 times more likely to be underweight.

According to 2019 Ethiopia Mini Demographic and Health Survey (EMDHS), 7% of children in Ethiopia are wasted, and 1% are severely wasted (below -3 SD) [18]. While our study identified higher prevalence of wasting in patients with CHD, which is 41.3%. In the same report of EMDHS, results show that 21% of all children are underweight (below-2 SD), while the prevalence is higher in our participants, 49.1%. The presence of such malnutrition will subject them to develop complication related to malnutrition and could also affect the outcome of the intervention, while the long term complications of malnutrition is not forgotten[7–10]. This research might help policy makers and/or nutritionists to come up with new strategies in the prevention and management of malnutrition in such patients who are at higher risk.

## 5. Limitations of the study

This study failed to assess an important contributing factor to malnutrition in CHD, that is CHF. This was not possible because study subjects were on outpatient follow up only.

The limited no of cyanotic patients included in the study also made comparing the two groups difficult (cyanotic vs acyanotic) and the number of patients with CHD and whose age is above ten years were few which makes difficult to draw any conclusion.

## 6. Conclusion

Malnutrition is a major problem in children with congenital heart disease. This indicates that proper and routine nutritional assessment should be part of the care while dealing with such patients and appropriate measures need to be undertaken this problem. Subsequent studies with larger sample size may strengthen the findings in this research and also the impact of malnutrition in such patients can further be studied in detail.

## Supporting information

**S1 File.**
(SAV)

## Acknowledgments

We are grateful to all who have been supporting in the pursuit of this study from its very beginning to reach into this level. All the comments we have been receiving from different committee members in the process were enriching and we are highly indebted to them.

## Author Contributions

**Conceptualization:** Temesgen Tsega, Azene Dessie.

**Data curation:** Tigist Tesfaye.

**Formal analysis:** Temesgen Tsega, Tigist Tesfaye.

**Methodology:** Temesgen Tsega, Azene Dessie, Tesfalem Teshome.

**Supervision:** Temesgen Tsega, Azene Dessie, Tesfalem Teshome.

**Validation:** Temesgen Tsega.

**Writing – original draft:** Tigist Tesfaye.

**Writing – review & editing:** Temesgen Tsega, Tigist Tesfaye.

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
