## [Decision Letter · Decision Letter 0]

9 Jul 2021

PONE-D-21-07948

Nutritional assessment and associated factors in children with congenital heart disease in Ethiopia.

PLOS ONE

Dear Dr. Desta,

Thank you for submitting your manuscript to PLOS ONE. After careful consideration, we feel that it has merit but does not fully meet PLOS ONE’s publication criteria as it currently stands. Therefore, we invite you to submit a revised version of the manuscript that addresses the points raised during the review process.

The reviewers were particularly concerned about the statistical analysis, which needs to be revised according to the suggestions of the reviewers, including, but not limited to, subgroup analyses as outlined in the reviewers' comments. In addition, a revised manuscript version needs to carefully follow the journal formatting guidelines. 

We look forward to receiving your revised manuscript.

Kind regards,

Laszlo Farkas, MD

Academic Editor

PLOS ONE

Journal Requirements:

Reviewers' comments:

Reviewer's Responses to Questions

**Comments to the Author**

1. Is the manuscript technically sound, and do the data support the conclusions?

Reviewer #1: Yes

Reviewer #2: Partly

2. Has the statistical analysis been performed appropriately and rigorously? 

Reviewer #1: Yes

Reviewer #2: No

3. Have the authors made all data underlying the findings in their manuscript fully available?

Reviewer #1: Yes

Reviewer #2: No

4. Is the manuscript presented in an intelligible fashion and written in standard English?

Reviewer #1: Yes

Reviewer #2: Yes

5. Review Comments to the Author

Reviewer #1: The authors conducted an interesting study with some sort of clinical importance. They assessed the impact of nutritional profile on children with CHF, considering a list of confounders that includes their socio-demographic data. Their statistical tools were sound and results supported their finding and presented in clean way.

Reviewer #2: Dear authors,

1- The current manuscript is written in a class education fasion and this fasion could not be accepted as journal paper. You should rewrite the manuscript and follow carefully the journal's guidelines for authors regarding proper manuscript structure and design.

2- The authors enrolled children from 1 year to 17 year old. Assessing nutritional status in a 2 year old child is not like assessing nutrional status in a 17 year old teenage. The age limit should be re considered.

3- The authors should make a separate and complete analyses using subgroup analyses for different age groups (i.e., <5 yr, from 5 to 12 yr, from 12 to 17 yr).

4- A comparison group composed of healthy children from the same region, same community and same age is needed. This is very important to measure properly the significance of changes in nutritional status.

5- In the methodology and study protocol, calculate the minimal sample size needed to conduct the study.

6- In the analysis, make adjustment for the relevant, significant and confounding variables before concluding the results.

7- The authors includes all types of congenital heart disease (I.e., simple defects and complex defect, treated/corrected and untreated/uncorrected). This point should be clear, revised, defended and properly discussed.

Regards

6. PLOS authors have the option to publish the peer review history of their article (what does this mean?). If published, this will include your full peer review and any attached files.

Reviewer #1: No

Reviewer #2: **Yes: **Rami Riziq Yousef Abumuaileq

---

## [Author Response · Author response to Decision Letter 0]

1 Oct 2021

Dear Reviewers, we would like to thank you very much for your precious time in revising the paper and for your invaluable comments to enrich the manuscript. 

We would like apologies and be thankful for the editorial team as well for our delayed responses as we have postponed the date for our response after informing the editorial team as we have been doing another research which is a continuation of this one that is assessing the impact of corrective intervention in malnourished children with CHD.

---

## [Decision Letter · Decision Letter 1]

9 Feb 2022

PONE-D-21-07948R1Nutritional Assessment and Associated Factors In Children with Congenital Heart Disease In Ethiopia.PLOS ONE

Dear Dr. Desta,

Thank you for submitting your manuscript to PLOS ONE. After careful consideration, we feel that it has merit but does not fully meet PLOS ONE’s publication criteria as it currently stands. Therefore, we invite you to submit a revised version of the manuscript that addresses the points raised during the review process.

Please address the comments raised by the reviewers, particularly related to manuscript formatting guidelines of the journal - please ensure that manuscript formatting follows journal guidelines. Also, clarify the results as requested by the reviewers.

We look forward to receiving your revised manuscript.

Kind regards,

Laszlo Farkas, MD

Academic Editor

PLOS ONE

Journal Requirements:

Reviewers' comments:

Reviewer's Responses to Questions

**Comments to the Author**

1. If the authors have adequately addressed your comments raised in a previous round of review and you feel that this manuscript is now acceptable for publication, you may indicate that here to bypass the “Comments to the Author” section, enter your conflict of interest statement in the “Confidential to Editor” section, and submit your "Accept" recommendation.

Reviewer #2: (No Response)

Reviewer #3: (No Response)

2. Is the manuscript technically sound, and do the data support the conclusions?

Reviewer #2: (No Response)

Reviewer #3: Yes

3. Has the statistical analysis been performed appropriately and rigorously? 

Reviewer #2: (No Response)

Reviewer #3: Yes

4. Have the authors made all data underlying the findings in their manuscript fully available?

Reviewer #2: Yes

Reviewer #3: Yes

5. Is the manuscript presented in an intelligible fashion and written in standard English?

Reviewer #2: (No Response)

Reviewer #3: No

6. Review Comments to the Author

Reviewer #2: The manuscript has been improved. We still have two comments:

1- In the discussion before the limitations section, the authors should highlight the clinical implications of the current study. These clinical implications should be clear and practical.

2- The authors should revise carefully the care checklist of the journal.

Regards

Reviewer #3: I congratulate authors to prove that malnutrition status of 208 pediatric CHD patients in Ethiopian population generally agrees with the reported populations. Applying these results enables to consider dietary or nutrient strategies to certain risk careering CHD patients.

However, I have some critiques authors should address:

Informed consent

It is shown as “patient’s oral consent” in ethic statement of cover letter. However in the paper text, it is shown as “written consent”. Please harmonize.

Format and structure of the paper

The paper does not align with a standard journal paper format. For example, rather long introduction with short discussion. Authors should reconcile.

• Introduction sections 1.1 to 1.3 are repetitive. These sections should be consolidated into one section.

• Authors should remove section 2. literature review. Then, the contents can be integrated into discussion or introduction section.

• Section 3. Objectives. Authors should not use bullet style format.

• Section 4.1. Irrelevant information about hospital. Please reconcile.

• Sections 4.2-4.4: Repetitive, please consolidate

• Section 4.10. Irrelevant, please remove

Abstract

• Children age between 5-10? Since Authors revised table. Same applies to conclusion, please update.

Methodology

Section 4.4. Authors need citation of a research from Egypt to determine sample size.

Results.

• Authors conducted bivariate analysis for certain set of variables followed by multivariate analysis. In the table 5.x, authors showed only three variables each without mentioning other variables. Did other variables meet statistically significant associations? Authors should clarify this in the text.

• For bivariate analysis, authors used both Acyanotic CHD and cyanotic CHD. I do not think there were control population. Does this have to be “CHD type”?

Discussion

• In your population or general CHD studies, Do patients who have PAH exhibit more severe condition? Do patients only have ASD exhibit less severe condition? Then, it makes sense for the associations with malnutrition. At least, you want to include in the discussion.

7. PLOS authors have the option to publish the peer review history of their article (what does this mean?). If published, this will include your full peer review and any attached files.

Reviewer #2: **Yes: **Rami Riziq Yousef Abumuaileq

Reviewer #3: No

---

## [Author Response · Author response to Decision Letter 1]

7 Mar 2022

I have uploaded my response as a separate file. Here again, I have copied my responses.

We would like to extend our gratitude for the reviewers and editors for your time and valuable comments. We have addressed all the comments in three pages and the response is written below each comment accordingly. All revised contents in the new manuscript are highlighted with light blue color. 

From one of the reviewers (comment #5) there was also a comment about the use of standard English. While revising the manuscript, we have tried to address some of the errors. We highlighted the corrected part in yellow. 

Thanks

Temesgen

Reviewer #2: 

The manuscript has been improved. We still have two comments:

1- In the discussion before the limitations section, the authors should highlight the clinical implications of the current study. These clinical implications should be clear and practical.

2- The authors should revise carefully the care checklist of the journal.

Response:

1- As a last paragraph in the discussion (just above “limitation of the study”) we have added a new paragraph which describes the clinical implication of our study. 

2- The manuscript is restructured as per PLOS ONE ‘manuscript organization’ guideline. So, we made corrections on the first page and the table of contents as some contents need to be either reshuffled or removed. The correction made in the manuscript are highlighted in light blue color. 

- Acronym and abbreviations - rewritten.

- The abstract page reduced to one.

- Abbreviations are removed from the abstract. 

- According to manuscript organization, the literature review and objective are removed while the objective in the abstract part has been left as it was.

- Acknowledgment added.

Reviewer #3: 

Informed consent

It is shown as “patient’s oral consent” in ethic statement of cover letter. However in the paper text, it is shown as “written consent”. Please harmonize.

Response: 

In the abstract there is a phrase stating about consent and it is corrected and rephrased same as the one mentioned in the ‘ethical consideration’ as ‘Data was collected from patient card and care givers of the children included in the study after obtaining their informed written consent using data inquiry sheet.’

I could not find a phrase stating about consent in the cover letter, which I have attached again. 

Reviewer #3:

Format and structure of the paper

The paper does not align with a standard journal paper format. For example, rather long introduction with short discussion. Authors should reconcile.

• Introduction sections 1.1 to 1.3 are repetitive. These sections should be consolidated into one section.

Response: We have minimized the introduction part by avoiding repetitive information. 

- Section 1.2 and 1.3 are minimized and specially 1.3 is rewritten again. 

Reviewer #3:

Format and structure of the paper Authors should remove section 2. literature review. Then, the contents can be integrated into discussion or introduction section.

Response: Literature review part is removed, and it was not even in the PLOS ONE journal manuscript organization format. 

• Section 3. Objectives. Authors should not use bullet style format.

Response: According to manuscript organization the objective is removed while it is left in the abstract part as it was.

• Section 4.1. Irrelevant information about hospital. Please reconcile.

Response: Some unnecessary phrases are removed. 

• Sections 4.2-4.4: Repetitive, please consolidate

Response: Re-written 

• Section 4.10. Irrelevant, please remove

Response: removed 

Abstract

• Children age between 5-10? Since Authors revised table. Same applies to conclusion, please update.

Response: corrected as per the comment. 

Methodology

Section 4.4. Authors need citation of a research from Egypt to determine sample size.

Response: Reference # 13. Hassan BA, Albanna EA, Morsy SM, Siam AG, Al Shafie MM, Elsaadany HF, Sherbiny HS, Shehab M, Grollmuss O. Nutritional Status in Children with Un-Operated Congenital Heart Disease: An Egyptian Center Experience. Front Pediatr. 2015 Jun 15;3:53. doi: 10.3389/fped.2015.00053. PMID: 26125014; PMCID: PMC4467172.

• Authors conducted bivariate analysis for certain set of variables followed by multivariate analysis. In the table 5.x, authors showed only three variables each without mentioning other variables. Did other variables meet statistically significant associations? Authors should clarify this in the text.

Response: Multivariable analysis was done to see if there is significant association of factors identified on bivariate analysis. We are mentioning here only those variables which showed associations. We put this statement in the result part above the table (highlighted in blue color).

• For bivariate analysis, authors used both Acyanotic CHD and cyanotic CHD. I do not think there were control population. Does this have to be “CHD type”?

Response: We used this quantitative analysis to compare two variables which are CHD types otherwise we don’t have a control group. 

Discussion

• In your population or general CHD studies, Do patients who have PAH exhibit more severe condition? Do patients only have ASD exhibit less severe condition? Then, it makes sense for the associations with malnutrition. At least, you want to include in the discussion.

Response: yes. Those patients with PAH are at higher risk of developing malnutrition, especially wasting and stunting as we mentioned it in the result Table 5b and 5c. 

The other finding in this study is the decreased risk of being underweight and stunted (70% (95% CI: 0.1-0.9)) in children with ASD than those CHD without ASD.

As per your comment we put the above statements in the discussion as well (page 18 and 19)

---

## [Decision Letter · Decision Letter 2]

29 Mar 2022

PONE-D-21-07948R2Nutritional Assessment and Associated Factors In Children with Congenital Heart Disease - EthiopiaPLOS ONE

Dear Dr. Desta,

Thank you for submitting your manuscript to PLOS ONE. After careful consideration, we feel that it has merit but does not fully meet PLOS ONE’s publication criteria as it currently stands. Therefore, we invite you to submit a revised version of the manuscript that addresses the points raised during the review process.

The reviewers found merit in your revised manuscript, but expressed remaining concerns on formatting and result reporting in the manuscript, which should be addressed.

We look forward to receiving your revised manuscript.

Kind regards,

Laszlo Farkas, MD

Academic Editor

PLOS ONE

Journal Requirements:

Reviewers' comments:

Reviewer's Responses to Questions

**Comments to the Author**

1. If the authors have adequately addressed your comments raised in a previous round of review and you feel that this manuscript is now acceptable for publication, you may indicate that here to bypass the “Comments to the Author” section, enter your conflict of interest statement in the “Confidential to Editor” section, and submit your "Accept" recommendation.

Reviewer #2: All comments have been addressed

Reviewer #3: (No Response)

2. Is the manuscript technically sound, and do the data support the conclusions?

Reviewer #2: Yes

Reviewer #3: Yes

3. Has the statistical analysis been performed appropriately and rigorously? 

Reviewer #2: Yes

Reviewer #3: Yes

4. Have the authors made all data underlying the findings in their manuscript fully available?

Reviewer #2: (No Response)

Reviewer #3: Yes

5. Is the manuscript presented in an intelligible fashion and written in standard English?

Reviewer #2: (No Response)

Reviewer #3: No

6. Review Comments to the Author

Reviewer #2: The authors have addressed our comments and the manuscript has been improved. The authors need to follow the journal's style

and follow the Care checklist.

Kindest regards

Reviewer #3: Authors have partly addressed my comments. Authors further need to address the comments below.

Informed consent

Please align to written consent from oral consent in Ethics Statement, which is in 5th page of the PDF manuscript (PONE-D-21-07948_R2)

Introduction:

Page 7 from line 8-12, I do not understand the wording here. Please reword using standard English.

Results:

Authors categorize acyanotic and cyanotic as CHD type in Table 2. Later, authors state both acyanotic CHD and cyanotic CHD have significant associations by bivariate analysis (last line of P15 and two more in P16 and 17). Author confirmed no control group was included. Please remove both “acyanotic CHD” and “cyanotic CHD”, and replace to “CHD type”.

Discussion:

P18, line 1-3, 11 and 20, Authors added parentheses and patient numbers. These are confusing with the citations. Please remove.

7. PLOS authors have the option to publish the peer review history of their article (what does this mean?). If published, this will include your full peer review and any attached files.

Reviewer #2: **Yes: **Rami Riziq Yousef Abumuaileq

Reviewer #3: No

---

## [Author Response · Author response to Decision Letter 2]

10 Apr 2022

Responses to reviewers 

Hello Dears, 

Thank you again for your time and continuous support. 

Responses for the specific comments are highlighted yellow in the revised manuscript. One of the comments from reviewers was the use of standard English. I have gone through the manuscript again and corrected many grammatical errors and words which are shaded with red. 

Review Comments to the Author

Reviewer #3: Authors have partly addressed my comments. Authors further need to address the comments below.

Informed consent

Please align to written consent from oral consent in Ethics Statement, which is in 5th page of the PDF manuscript (PONE-D-21-07948_R2)

Response:

Corrected as ‘written consent’. 

Introduction:

Page 7 from line 8-12, I do not understand the wording here. Please reword using standard English.

Response:

Addressed accordingly, is highlighted in yellow. 

Results:

Authors categorize acyanotic and cyanotic as CHD type in Table 2. Later, authors state both acyanotic CHD and cyanotic CHD have significant associations by bivariate analysis (last line of P15 and two more in P16 and 17). Author confirmed no control group was included. Please remove both “acyanotic CHD” and “cyanotic CHD”, and replace to “CHD type”.

Response: Corrected as per the comment; in the revised manuscript page 16 and 17 

Discussion:

Comment: P18, line 1-3, 11 and 20, Authors added parentheses and patient numbers. These are confusing with the citations. Please remove.

Response:

Removed as per the comment. page 14 and 15 highlighted with red. 

The last sentence on page 6 is removed because it does not explain specifically why Infants and children are more likely to suffer from poor nutrition compared to adults.

---

## [Decision Letter · Decision Letter 3]

24 May 2022

Nutritional Assessment and Associated Factors In Children with Congenital Heart Disease - Ethiopia

PONE-D-21-07948R3

Dear Dr. Desta,

We’re pleased to inform you that your manuscript has been judged scientifically suitable for publication and will be formally accepted for publication once it meets all outstanding technical requirements.

Kind regards,

Laszlo Farkas, MD

Academic Editor

PLOS ONE

Additional Editor Comments (optional):

Reviewers' comments:

Reviewer's Responses to Questions

**Comments to the Author**

1. If the authors have adequately addressed your comments raised in a previous round of review and you feel that this manuscript is now acceptable for publication, you may indicate that here to bypass the “Comments to the Author” section, enter your conflict of interest statement in the “Confidential to Editor” section, and submit your "Accept" recommendation.

Reviewer #3: All comments have been addressed

2. Is the manuscript technically sound, and do the data support the conclusions?

Reviewer #3: Yes

3. Has the statistical analysis been performed appropriately and rigorously? 

Reviewer #3: Yes

4. Have the authors made all data underlying the findings in their manuscript fully available?

Reviewer #3: Yes

5. Is the manuscript presented in an intelligible fashion and written in standard English?

Reviewer #3: Yes

6. Review Comments to the Author

Reviewer #3: I congratulate authors to have a manuscript met PLOS ONE standards. All comments are addressed properly.

7. PLOS authors have the option to publish the peer review history of their article (what does this mean?). If published, this will include your full peer review and any attached files.

Reviewer #3: No

---

## [Editor Report · Acceptance letter]

2 Jun 2022

PONE-D-21-07948R3 

*Nutritional Assessment and Associated Factors In Children with Congenital Heart Disease - Ethiopia*

Dear Dr. Desta:

I'm pleased to inform you that your manuscript has been deemed suitable for publication in PLOS ONE. Congratulations! Your manuscript is now with our production department. 

Kind regards, 

on behalf of

Dr. Laszlo Farkas 

Academic Editor

PLOS ONE